# Digital Bicycling Planning: A Systematic Literature Review of Data-Driven Approaches

**Parisa Zare** [1,*], **Christopher Pettit** [1], **Simone Leao** [1] **and Ori Gudes** [2]

1   School of Arts, Design & Architecture, University of New South Wales, Sydney, NSW 2052, Australia
2   School of Population Health, University of New South Wales, Sydney, NSW 2052, Australia
*   Correspondence: p.zare@unsw.edu.au; Tel.: +61-4-5261-6768

**Abstract:** To increase the amount of bicycling as a mode of transport, many countries are developing placed based bicycling plans and strategies. However, this approach necessitates considering a fine-scale mapping of bicycling patterns and a detailed description of urban spaces. The rise of new data and technologies offers much promise to planners and researchers to access diverse and richer sources of information to optimise the bicycling network design. This review aims to comprehensively examine the role of data and technology in bicycling planning, historical changes in using data-driven approaches, and current domains in the existing body of research in bicycling planning from 1990 to 2021. For this, a systematic literature review has been conducted according to PRISMA framework. A total number of 1022 studies was analysed and synthesised with the VOS Viewer and CiteSpace platforms. Upon completing the review, we extracted the most-used datasets, tools, and methodologies. The results of the systematic review reveal three evolutionary phases in using data-driven approaches from 1990 to 1999, 2000 to 2009, and 2010 to 2021. In addition, we identified six knowledge domains in using data-driven approaches in bicycling planning that is (i) smart city, (ii) infrastructure, (iii) built environment, (iv) decision making, (v) people, and (vi) safety.

**Keywords:** bicycling planning; data-driven approaches; systematic literature review

## 1. Introduction

Cities worldwide are increasingly investing in developing their bicycling infrastructure. However, due to limited active budgets in many cities, it is essential to optimise a comprehensive network for bicycling [1]. Decisions about where to prioritise investment in cities have generated a need for more information and data analysis methods to estimate bicycling demand more efficiently [2].

Researchers and practitioners traditionally obtained data and information on bicycling from traffic counts, travel surveys, and field observation. With the rise of data-driven approaches coupled with the crowd-sourcing and open data revolution, more detailed data and analytical tools to examine them are available [3]. Using Global Positioning Systems (GPS) enables recording bicyclist movement and travel routes via crowd-sourcing applications embedded in smartphones. In addition, many cities are sharing their spatial data, such as bicycling infrastructure, land use, and vehicle traffic data, publicly available as part of the emerging open data movement [4]. In the current literature, few studies tried to identify, classify, and analyse some aspects of data-driven approaches, such as emerging datasets such as crowd-sourced data and their applications in pedestrian and bicycle monitoring [3,5,6]. In this review, the authors extensively examine the existing body of research in bicycling planning to discuss the role of data-driven approaches that includes datasets, tools, and methodologies used in bicycling planning, understand the historical change in using data-driven approaches from 1990–2021, and map the current domains in the existing body of research in bicycling planning. In this context, the authors pose the following research question:

- What are data-driven approaches, including datasets, tools, and methodologies, applied in bicycling planning?
- What is the evolving role of data and technology in bicycling planning research from 1990 to 2021?

This study used a systematic literature review methodology and quantitative and qualitative analysis methods to address the research questions. The structure of this manuscript is as follows: Section 2 presents the methods developed and applied to do a systematic literature review and analysis methodology. Section 3 describes the results of the analysis based on the methods used. Section 4 discusses the findings and research domains. Finally, Section 5 outlines the conclusion and future research directions.

## 2. Methodology and Analysis

Systematic literature reviews help collect and find all related studies based on the defined inclusion and exclusion criteria to answer the research questions by applying scientific, replicable and transparent procedures [7]. This review is aligned with the Preferred Reporting Items for Systematic Reviews and Meta-Analyses (PRISMA) guidelines. According to PRISMA guidelines (Supplementary Materials), a framework for doing a literature review includes three main phases: (1) identification (2) screening and eligibility criteria and (3) data extraction [8]. This methodology has rarely been used in urban studies. Due to the complexity of most of the concepts in this field, there is a need to have more specified definitions of keywords. Therefore, to address this issue, the methodological approach used in this review is combined with a systematic literature review model initially proposed in [9]. In the proposed model by [9], systematic literature review phases are (1) definition of the appropriate scope, (2) conceptualisation of the topic, (3) literature search, (4) literature analysis and synthesis, and (5) research agenda. Figure 1 represents the systematic framework and the workflow applied in this literature review. We undertook a broad conceptualisation of the topic to do the literature search efficiently. After that, we systematically reviewed, analysed, and synthesised the selected studies. The last step was the research agenda, which aimed to present core trends in using data-driven methods in the literature on bicycling planning.

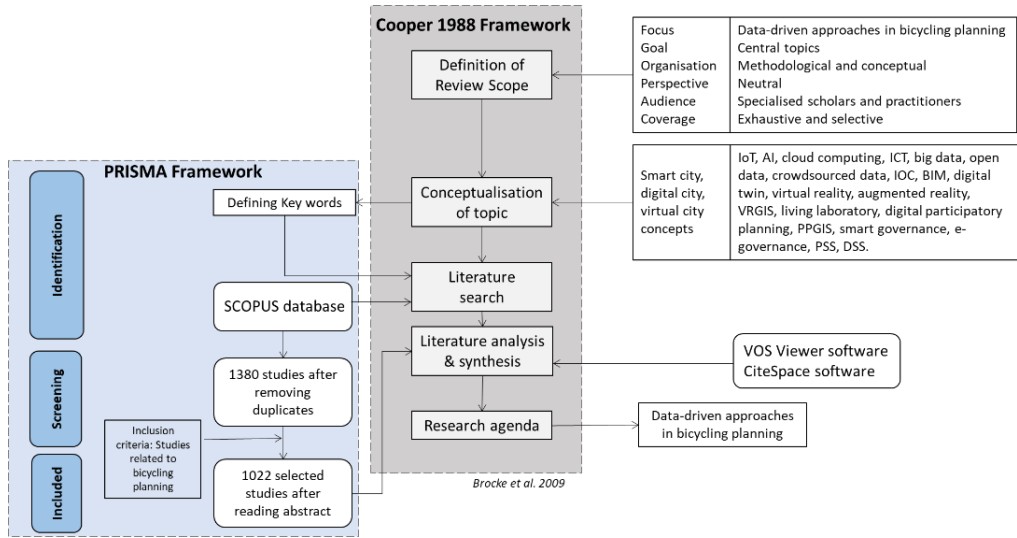

**Figure 1.** Framework for the literature review on data-driven approaches in bicycling planning research.

### 2.1. Definition of the Literature Review Scope

Identification of the related literature requires defining the scope of the topic. A taxonomy suggested by [10] was used to determine the extent of this review. Cooper's taxonomy method includes six characteristics: focus, goal, organisation, perspective, audience, and coverage. In this review, the (1) focus is on methodologies and approaches; (2) the goal

of this review is to analyse past studies and identify the core topics in using data-driven approaches in bicycling planning; (3) the organisation of this review is methodological and conceptual order that is to group the applied methods; (4) this review has a neutral representation perspective; (5) the audience are specialised scholars and practitioners in bicycling planning; and (6) this review is exhaustively cover studies related to the topic.

### 2.2. Conceptualisation of the Topic

In this review to identify the core concept, we began to study the data-driven approaches in bicycling planning by examining the literature and following a primary search on google scholar, TRID, Web of Science and Scopus databases. Keywords for this comprehensive review are "Bicycling" OR "Bicycle" OR "Cycling" OR "Bike" AND "Data AND Model". We reviewed more than 200 studies comprehensively to have a primary idea about the applied methods and technologies (particularly [11]). Exploring these studies, we found key topics and concepts that can represent data-driven approaches in the bicycling planning literature, such as Smart City, Internet of Things (IoTs), Artificial Intelligence (A.I), Cloud Computing, Information and communication technologies (ICT), Intelligent Operation Centre (IOC), Building Information Modelling (BIM), Digital Twin, Virtual Reality (VR), Augmented Reality (AR), VRGIS, Living Laboratory, Digital Participatory Planning, Participatory Geographic Information Systems (GIS), Smart Governance (Table A1 in Appendix A represents these concepts and keywords and lists their definitions.). It is noticeable that some of these concepts have meanings that overlap (such as digital twins and smart city). In addition, it can be concluded that data-driven is a broad concept, and there is no unique definition. For these reasons, all these concepts identified in the conceptualisation step are included in the next step of the literature search.

### 2.3. Literature Search

This phase includes selecting the databases, keywords, and backward and forward searches [12]. First, we choose Scopus among the current databases. Scopus enables researchers to access the peer-reviewed literature dataset by Elsevier and Science Direct [13]. Second, key concepts in the previous step are used for defining keywords to create a better framework in this systematic review (Appendix B includes the search criteria). This search was limited to English engineering, computer science, environmental science, social science, mathematics, decision sciences, business and management, arts and humanities, psychology, or health studies in the Scopus database. In both search engines, selected studies are from 1990 to 2021 to clarify the evolutionary trend in using data-driven approaches in bicycling planning (according to preliminary research in the datasets, this is the start of using census data, crash data, and tools such as geographic information systems). This search results in 1380 in the Scopus search engine in December 2021. According to the defined literature search methodology, there is a need to apply backward and forward searches in the next step. In this review, the number of selected studies is considered an appropriate number to investigate the data-driven approaches in bicycling planning.

In the last step, the authors conduct an evaluation to limit the number of identified studies. To do this, 358 unrelated studies were removed manually by reading the studies' abstracts (For instance [14] used surveys to model the bicyclists behaviour rather than using data-driven methots or in a study by [15], characteristics of bicyclsing behaviour was evaluated using video images of bicycle traffics). The selected 1022 studies' records are characterised by authors' names, publication years, titles, journals, cities (or countries), keywords, and abstracts. The final database is the primary resource in this study for data mining to achieve the research objectives.

### 2.4. Literature Analysis and Synthesis

Based on the (Brocke et al. 2009 [9]) approach, the fourth phase synthesises the collected studies and organises them for a systematic analysis. Researchers developed many tools to analyse a systematic literature review with this approach, such as SCIMAT,

BibExcel, Ucinet, VOS Viewer, NVIVO, and CiteSpace. In this review, VOS Viewer and CitesSpace 5.6.R2 were used. VOS Viewer is specialised software for cluster analysis and visualisation [16]. CiteSpace is suitable for evolutionary assessment based on the co-occurrence of keywords [17]. In this systematic review, the co-occurrence of keywords (counting of paired data in a defined unit) is the core of the analysis methods. Keywords co-occurrence analysis can help clarify and identify the core topics in the dataset. This co-occurrence analysis is primarily based on the full counting of keywords in the title, abstract, and keywords of studies. In the full counting method, the occurrence of items represents the total number of occurrences of a keyword in all studies [18]. As a result, the core analysis methods that the authors used to investigate the studies are (1) cluster analysis to analyse the main research contents and trends using VOS Viewer and (2) temporal analysis of the research about the data-driven approach using CiteSpace.

### 2.4.1. Cluster Analysis

Cluster analysis is a data mining method for identifying and analysing the main keywords, contents, topics, and interrelationships [19]. In this context, identifying research clusters in the literature help to organise a large number of studies into a substantial number of groups and extract information from each group. VOS Viewer performs this cluster analysis by extracting the co-occurrences of items. Clusters in VOS Viewer are a group of items included in a map. Items can be publications, researchers, or terms. In this review, items represent keywords in the dataset. These items are usually linked together based on the relation between two items (the links show the co-occurrence between terms). The strength of links between terms indicates the number of publications in which two terms occur together. An item can only belong to one cluster; therefore, the result will be a non-overlapping set of clusters. This software labels clusters by numbers starting from cluster number 1. Items can have different attributes and characteristics. If an item belongs to one cluster, cluster numbers and specifications are examples of its attributes. The most important attribute of each item is its weight which means its importance (weight is non-negative value, and an item with a higher weight is more important, and in visualisation, it is indicated more prominently). The normalisation method in the VOS Viewer is association strength. This probabilistic measure represents the ratio of the observed number of co-occurrences to their expected number of co-occurrences [20].

### 2.4.2. Temporal Analysis

Temporal analysis reports patterns in using keywords based on time. Items used for text processing in this analysis are title, abstract, and keywords, and the type of the nodes are keywords. This temporal analysis has a 5-year range in chronological order to represent the evolution of the keywords that occurred more than ten times from 1990 to 2021. We also distinguished topics and methods.

### 2.5. The Research Agenda

The result of a literature review should be a research agenda that include more sophisticated and insightful questions for extending the concept and literature review [21]. This research agenda includes a state of data-driven approaches in bicycling planning and a conceptual framework that deepens the core contents of methods and tools in the selected studies that will be presented in Section 4.

## 3. Results

This section reports the systematic literature review analysis results focusing on clustering main knowledge domains and evolutionary trends and in using data-driven approaches in bicycling planning research. Figure 2 represents the network visualisation of items and links in the selected papers using the VOS Viewer platform based on a minimum of twenty occurrences (frequency of keywords in the dataset). In this visualisation, 82 out of 8566 keywords are identified and clustered. A colour represents each cluster. In this

visualisation, 300 links have been represented, indicating the strongest relations between items using co-occurrence.

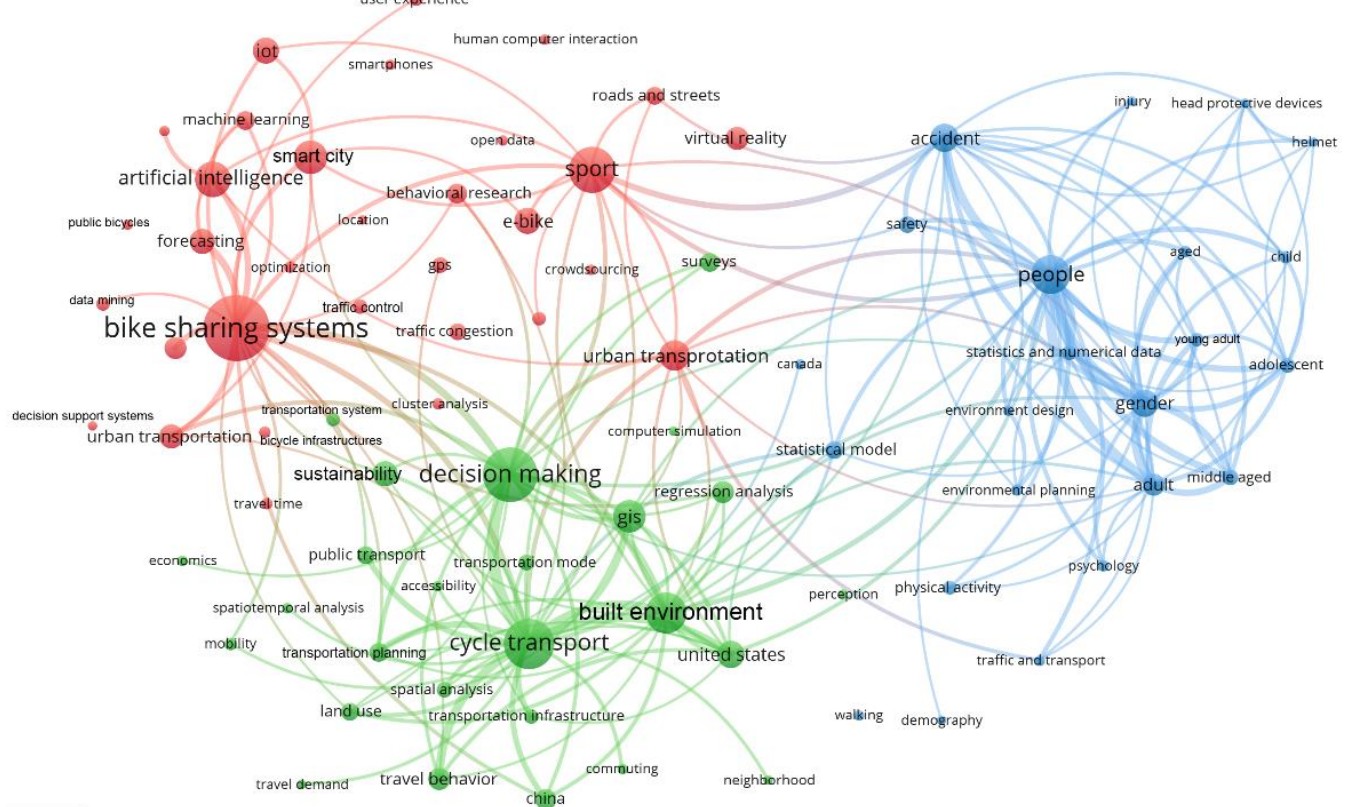

**Figure 2.** Co-occurrence of 82 first keywords in the selected studies' keywords based on the minimum occurrence of 20 among 8462: Cluster 1: Smart city and infrastructure (red nodes), Cluster 2: Built environment and decision making (green nodes), Cluster 3: People and safety (blue nodes).

In the red cluster, the most repeated words are "sharing system", "smart city", and "artificial intelligence". The green cluster includes words such as "decision-making", "built environment", and "travel behaviour". In the blue cluster, the most repeated words are "human", "safety", and "traffic accident". Overall, based on the most repeated words in each cluster, these clusters can be labelled as cluster 1: Smart city and infrastructure, cluster 2: Built environment and decision making, and cluster 3: People and safety.

The result of the temporal analysis is presented in Table 1, which shows keywords that occurred more than ten times from 1990 to 2021. To make this analysis consistent with the cluster analysis, we use the result of the cluster analysis to group the keywords. According to the analysis results, there are no keywords in the 1990–1994 time slot in Smart city and decision-making clusters. Keywords in this time slot are concerned with safety and demographics. Topics that emerged in the 1995–1999 time slot are "computer simulation", "mathematical modelling", "head protective device" ("helmet"), "adults", "behavioural research", and "regression analysis". New topics that emerged in the 2000–2004 time slot include keywords such as "virtual reality", "accident prevention", "physical activity", "GIS", and "geographic information systems". The most repeated keyword in the 2005–2009 time slot is "cycle transport", "built environment", and "decision making".

**Table 1.** Co-occurrence of keywords in the 5-year range in chronological order from 1990 to 2021.

| Evolutionary Phase | | Phase 1 | | | |
|---|---|---|---|---|---|
| cluster | type | 1990–1994 | | 1995–1999 | |
| Smart city and infrastructure cluster | Topics | | | Urban planning | 29 |
| | Methods and tools | | | Computer simulation | 20 |
| | | | | Mathematical model | 15 |
| | | | | simulators | 15 |
| Built environment and decision making | Topics | | | United states | 80 |
| | | | | Behavioural research | 49 |
| | Methods and tools | | | Regression analysis | 56 |
| People and safety cluster | Topics | People | 132 | Head protective device | 20 |
| | | Gender | 145 | Helmet | 22 |
| | | Child | 28 | Adult | 59 |
| | | Adolescent | 35 | | |
| | | Injury | 28 | | |
| | | Accident | 64 | | |
| | Methods and tools | Comparative study | 15 | biomechanics | 10 |

| Evolutionary Phase | | Phase 2 | | | |
|---|---|---|---|---|---|
| cluster | type | 2000–2004 | | 2005–2009 | |
| Smart city and infrastructure cluster | Topics | China | 54 | Motor transportation | 31 |
| | | Automation | 15 | Electric bicycle | 29 |
| | Methods and tools | Virtual reality | 61 | Navigation | 10 |
| | | | | visualisation | 10 |
| | | | | Mobile device | 10 |
| | | | | User interface | 17 |
| | | | | Human computer interaction | 20 |
| Built environment and decision making | Topics | Environmental planning | 15 | Decision making | 191 |
| | | | | Cycle transport | 174 |
| | | | | Built environment | 130 |
| | | | | Planning | 11 |
| | Methods and tools | Geographic Information System | 63 | Socio economic factor | 14 |
| People and safety cluster | Topics | Accident prevention | 32 | Pedestrian safety | 15 |
| | | Traffic accident | 43 | | |
| | | Aged | 25 | | |
| | | Physical activity | 32 | | |
| | | Middled aged | 39 | | |
| | | Walking | 21 | | |
| | | Active transportation | 10 | | |
| | Methods and tools | Auditing instrument | 10 | Statistical model | 20 |

| Evolutionary Phase | | Phase 3 | | | | | |
|---|---|---|---|---|---|---|---|
| cluster | type | 2010–2014 | | 2015–2019 | | 2020–2021 | |
| Smart city and infrastructure cluster | Topics | Bicycle infrastructure | 34 | Bike sharing system | 169 | Dock less bike sharing | 23 |
| | | Bicycling sharing system | 84 | Sport | 153 | Demand prediction | 22 |
| | | | 27 | Deep learning | 20 | Sharing economy | 25 |
| | | Design | 35 | E-bike & electric bike | 44 | Urban mobility | 15 |
| | | Intelligent systems | 34 | Public bicycle | 24 | Demand modelling | 6 |
| | | | 116 | Smart city | 74 | Accurate prediction | 6 |
| | | Traffic control | 15 | | | Emission control | 5 |
| | | Sharing system | | | | K mean clustering | 5 |
| | | Energy utilization | | | | Service quality | 5 |
| | | | | | | Weather | 5 |
| | | | | | | Time series | 5 |
| | | | | | | Micro mobility | 5 |

**Table 1.** *Cont.*

| Evolutionary Phase | | Phase 3 | | | | | |
|---|---|---|---|---|---|---|---|
| cluster | type | 2010–2014 | | 2015–2019 | | 2020–2021 | |
| | Methods and tools | Artificial intelligence | 77 | User experience | 30 | Data sharing | 20 |
| | | Augmented reality | | Ubiquitous computing | 18 | Integer programming | 10 |
| | | Internet of things | 20 | Optimization | 24 | | |
| | | Forecasting | 64 | Open data and datum | 35 | | |
| | | Data set | 72 | Machine learning | 30 | | |
| | | Data collection | 20 | Learning system | 28 | | |
| | | Information | 15 | IOT | 38 | | |
| | | management | 16 | Data mining | 30 | | |
| | | Global positioning system | 28 | Cluster analysis | 23 | | |
| | | Travel time | | Big data | 17 | | |
| | | Smartphone | 24 | Agent based modelling | 59 | | |
| | | Decision | 22 | Data driven approach | 16 | | |
| | | support system | 22 | Demand prediction | 16 | | |
| Built environment and decision making | Topics | Land use | 45 | Sustainability | 28 | Environmental benefit | 5 |
| | | Neighbourhood | 21 | Accessibility | 21 | Housing | 5 |
| | | Road and street | 43 | | | | |
| | | Sustainable development | 47 | | | | |
| | | Transportation infrastructure | 37 | | | | |
| | | Transportation mode | 38 | | | | |
| | | Transportation planning | 47 | | | | |
| | | Urban area | 31 | | | | |
| | | Urban transportation | 67 | | | | |
| | | Economics | 22 | | | | |
| | | Transportation system | 32 | | | | |
| | | Investment | 28 | | | | |
| | | Commuting | 24 | | | | |
| | | Electric vehicle | 42 | | | | |
| | Methods and tools | Survey | 52 | Spatiotemporal analysis | 21 | Empirical analysis | 15 |
| | | Spatial analysis | 38 | Information Systems | 15 | STRAVA | 5 |
| | | Numerical model | 25 | Crowd-sourcing | 24 | | |
| | | Modelling | 17 | Cost benefit analysis | 15 | | |
| | | Level of service | 15 | Travel demand | 17 | | |
| | | | | Public policy | 15 | | |
| People and safety cluster | Topics | Demography | 20 | Safety engineering | 20 | Household income | 6 |
| | | Environment design | 22 | Air quality | 20 | | |
| | | Environmental planning | 26 | | | | |
| | | Travel behaviour | 65 | | | | |
| | | Young adult | 32 | | | | |
| | | Safety | 41 | | | | |
| | | Traffic congestion | 41 | | | | |
| | | Psychological aspect | 32 | | | | |
| | | Statistical model | 34 | | | | |
| | | Perception | 20 | | | | |
| | | Student | 12 | | | | |

**Table 1.** *Cont.*

| Evolutionary Phase | | Phase 3 | | | | |
|---|---|---|---|---|---|---|
| cluster | type | 2010–2014 | | 2015–2019 | | 2020–2021 |
| Methods and tools | | Population statistics | 33 | Survey and questionnaire | 10 | Human experiment | 5 |
| | | Controlled study | 17 | Questionnaire | 20 | |
| | | Cross sectional studies | 13 | Risk assessment | 12 | |
| | | Statistics and numerical model | 34 | | | |
| | | Safety device | 10 | | | |
| | | Risk factor | 10 | | | |

With the rapid development of "information technologies" from 2010–2014, bicycling planning researchers started to use techniques such as "artificial intelligence", "augmented reality", and "internet of things" to understand "travel behaviour" for a more "sustainable development". This trend includes introducing and applying "decision support systems" in "urban transportation" and "spatial analysis". Also, some of the most repeated keywords are "bicycling sharing system", "bicycling infrastructure", and "sharing systems".

The time slot from 2015 to 2019 represents the application of new methods such as "data mining", "machine learning", and "cluster analysis" methods that enables "optimisation" of the "design" of bicycling infrastructure network. Most repeated keywords are in the smart city and infrastructure cluster in this time slot. This trend applies to the time slot from 2020–2021 too. Although the frequency of keywords in the 2020–2021 time slot is less than others, there is an increasing trend in using keywords in smart city and infrastructure cluster. According to this temporal analysis, we propose to consider three phases ((i) from 1990 to 2000, (ii) from 2000 to 2010, and (iii) from 2010 to 2021) of using data-driven methods in bicycling planning in the following sections (refer to Table 1).

## 4. Discussion

As discussed in the previous section, there is an evolving set of topics, methods, and tools in using data-driven approaches in bicycling planning. Therefore, in the rest of this section, we present a clear framework for these topics, methods, and tools.

### 4.1. Knowledge Domains in Using Data-Driven Approaches

The cluster analysis results in the previous section show intersecting results of several research themes in using data-driven approaches in bicycling planning. However, these clusters intersect and reflect broad knowledge domains. Therefore, we divided each cluster into two knowledge domains (Figure 3 shows the results). The rest of this section is a discussion of these knowledge domains.

The first cluster is smart city and infrastructure, divided into two sub-categories (1) smart city and virtual reality and (2) bicycling infrastructure and bike-sharing systems. With the advent of the smart city concept, bicycling planning is not bound to traditional methods. Nowadays, it is possible to build a virtual environment to evaluate how bicyclists might behave in urban spaces using VR, AI, and IoTs [22]. Some of the studies in this review applied technologies that enable simulation of the bicyclist behaviour using modelling techniques such as multinomial logit models and experimental analysis (e.g., [23]), agent-based modelling (e.g., [24]), and locomotion (e.g., [22]).

According to the second sub-category, mobile devices and participatory health sensing are growing trends in collecting health and activity data using applications for recording bicycling and walking or other physical activities like iBike, Strava, RiderLog, and MapMyRIde [25]. They have different formats and functions, but they can gather a large amount of data (crowd-sourced data) on user activities [26]. In addition, there has been a sharp increase in the popularity and prevalence of shared bike studies in the last two decades. However, shared bike systems have been around for more than fifty years [27]. The traces of the shared bikes systems can be used to examine the bicyclists' behaviour in the built

environment. For instance [28] used the traces of shared bike trips to reveal the spatial patterns of urban spaces attractiveness. Cargo bikes are other recent popular topic in this sub-category for transportation of last mile goods deliveries in urban areas [29,30]. The most common analysis methods and approaches in this category that has been used in the reviewed studies are learning algorithms (e.g., [31]), cluster analysis (e.g., [32]), demand analysis (e.g., [33]), linear or non-linear regression analysis (e.g., [34]), and GIS analysis (e.g., [35]).

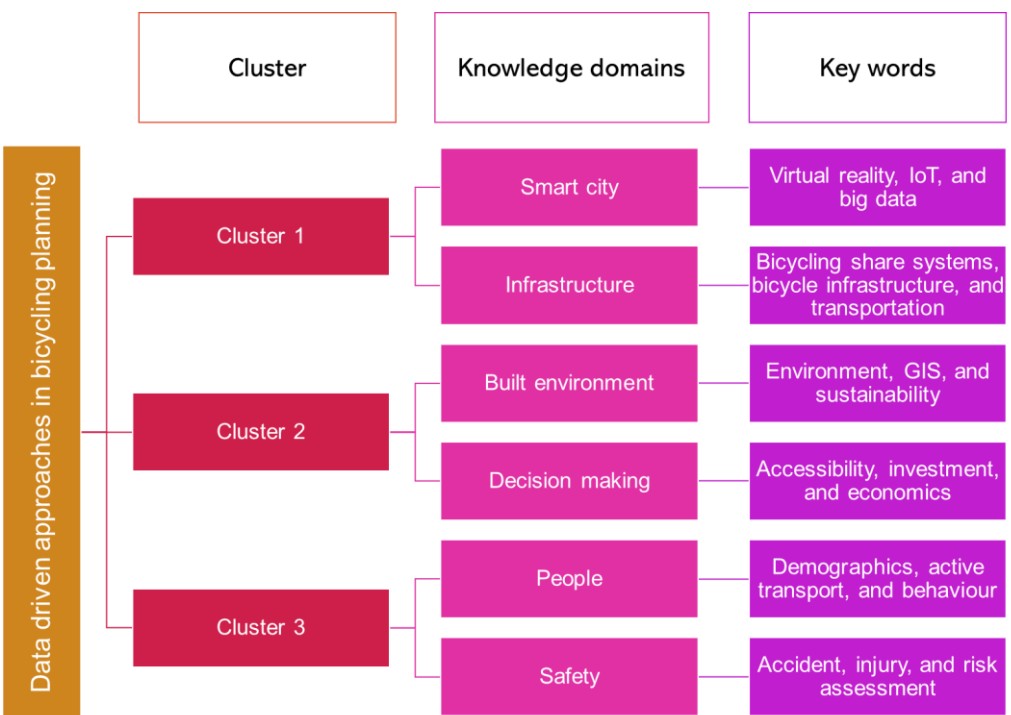

**Figure 3.** Knowledge domain categories.

The second cluster in this review is the built environment and decision-making divided into two smaller sub-categories of (1) built environment and travel behaviour and (2) decision-making and planning. According to the first sub-category, bicycling as a desirable activity in cities can be affected by attributes of the built environment [36]. The top ten built environment attributes that affect bicycling behaviour, according to the reviewed studies are:

- Bicycling infrastructure (e.g., physically separated bike path, painted cycleway, and shared path) (e.g., [37])
- Land use (e.g., land use entropy, land use mix, and primary destinations density) (e.g., [38])
- Network characterises (e.g., connectivity, road segment length, and accessibility) (e.g., [39])
- Personal and household attributes (e.g., age, gender, number of motor vehicles, and income) (e.g., [40])
- Distance between trip origin and destination (e.g., [41])
- Topography (e.g., slope and elevation) (e.g., [42])
- Density (e.g., population, residential, and employment) (e.g., [43])
- Public transport and vehicle travel costs (e.g., [44])
- Vehicle traffic attributes (e.g., volume and speed) (e.g., [45])
- Weather (e.g., rainfall, temperature, wind, and speed) (e.g., [46])

The second sub-category in the second cluster is decision-making and planning. According to the reviewed studies, accessibility is one of the crucial concepts in decision-making in the context of modelling land value and travel behaviour to examine economic

development using evaluation tools and methods (e.g., [47,48]). Other concepts are Bicycling Decision Support Systems (DSS) or Planning Support Systems (PSS) that represent the increasing role of digital technology in using data-driven approaches in the decision-making and policy-setting process (e.g., [49–52]). Table 2 outlines a summary of some of these tools and their application (such as estimating future demand, prioritising facility investment, and shared bike planning) in the bicycling planning process.

**Table 2.** Summary of some of the PSS used in bicycling planning (adapted with [53]).

| Tool | Country | Application | Output Format | Factors and Variables Included | Developer |
|---|---|---|---|---|---|
| Regression model as a prediction tool | England and Wales | Estimate the proportion of bicycling trips to work | Static model | Physical, transport, and socio-economic factors | [41] |
| Usage intensity index | Brazil | Prioritise key paths for bicycling | GIS-based | Attitudinal opinions, origin-destination, and socio-economic factors | [54] |
| Prioritisation index | Canada | Prioritise the location for bicycling facilities | GIS-based (grid cells) | Preferred routes, bicycling collision data, and bicycling network | [55] |
| Facility Demand | USA | Estimate future walking and bicycling | GIS-based | Bicyclists' characteristics, accessibility, land use, and transport factors | [56] |
| Prioritisation index | Canada | Prioritise the investment in bicycling facilities | GIS-based (grid cells) | Slope, length, origin-destination, preferred routes, and connectivity of the bicycling network | [49] |
| Heat map | England | Planning bicycling network | GIS-based (grid cells) | Observed trip characteristics like trip distance, bicyclists' characteristics, and time of the day | (Transport for London, 2010) |
| The propensity to cycle tool | England | Prioritise the investment | GIS-based | Origin-destination, gender composition, mortality rate, distance, and hilliness | [53] |
| Agent-Based Modelling (ABM) | USA | Support decision-making for investment | Simulation, GIS-based | Socio-demographic data and bicycling infrastructure | [37] |
| | Hong Kong | Support the shared bike system | Simulation, GIS-based | Bicyclists' characteristics, bike count, time, distance, duration, origin-destination, bicycling infrastructure, and speed | [50] |
| Cycling Measures Selector (CMS) | Portugal | Support stakeholders in defining cycling strategies | Web-based | Bicycling infrastructure, bicyclists' attitudes and behaviours, and land use | [51] |

The third cluster in this review is people and safety, divided into smaller sub-categories of (1) people and demographics and (2) safety and accident prevention. Reviewed studies in the first sub-category tried to examine the level of physical activity according to different demographic variables such as age and gender (e.g., [40]). Also, there are studies trying to understand the psychological and social aspects of bicycling (e.g., [52]). With the availability of rich online resources in people's opinions, such as reviews on websites and social media commentary, a new trend in research has emerged that attempts to find out the sentiments and behaviour of bicyclists in the urban environment (e.g., [57]).

The second sub-category in the third cluster is safety and accident prevention. Safety is one of the significant barriers to choosing to bicycle as a mode of transport, especially for short-distance trips. Therefore, researchers have examined how to provide safe paths and networks for bicyclists. Some of the topics and strategies in the reviewed studies are: introducing laws and regulations to oblige bicyclists to use helmets (e.g., [58]), assessing motor vehicle volume and road characteristics such as speed limit on bicyclist risk injury (e.g., [45]), and provisioning protected intersection design elements (e.g., [59]).

Datasets and Methods

Researchers and practitioners have used a range of open and crowd-sourced datasets to have a data-driven approach to bicycling planning. The most used datasets are census data (e.g., [60]), bicycle infrastructure network data (e.g., [61]), land use data (e.g., [62]), shared bike data (e.g., [63]), topography data (e.g., [39]), weather data (e.g., [46]), crash statistics data (e.g., [64]), General Transit Feed Specification (GTFS) data (e.g., [65]), and Open Street Map data (e.g., [66]). Table 3 illustrates the detail of these datasets and their usage according to the main knowledge domains in Section 4.1 and the evolutionary phases in Section 2.4.2.

**Table 3.** Summary of most-used datasets and methods.

| Evolutionary Phase | Dataset | Possible Applications | Smart City and Infrastructure Cluster | Built Environment and Decision Making | People and Safety |
|---|---|---|---|---|---|
| 1–3 | Census data | Travel mode choice and simulation of bicyclist behaviour | ✓✓ | ✓✓ | ✓✓ |
| 1–3 | Household travel survey | Transport mode choice and simulation of bicyclist behaviour | ✓ | ✓✓ | ✓✓ |
| 1–3 | Journey to work | travel mode choice and origins and possible destinations | ✓ | ✓✓ | ✓✓ |
| 1–3 | GPS survey data | Geographic location analysis, route choice analysis, and simulation of bicyclist behaviour | ✓ | ✓✓ | ✓ |
| 1–3 | Bicycling statistics, bike count data | Estimation of future demand, safety analysis, and simulation of bicyclist behaviour | ✓ | ✓✓✓ | ✓✓✓ |
| 1–3 | Crashes, accidents and injuries data | Safety analysis | ✓ | ✓ | ✓✓✓✓ |

**Table 3.** *Cont.*

| Evolutionary Phase | Dataset | Possible Applications | Smart City and Infrastructure Cluster | Built Environment and Decision Making | People and Safety |
|---|---|---|---|---|---|
| 1–3 | Bicycle infrastructure Network data | Built environment suitability analysis, safety analysis, and simulation of bicyclist behaviour | ✓✓ | ✓✓✓✓ | ✓✓✓✓ |
| 1–3 | Annual & Daily Traffic data | Built environment suitability analysis and safety analysis, | ✓ | ✓✓ | ✓✓ |
| 1–3 | Weather data | Estimation of future demand and simulation of bicyclist behaviour | ✓✓ | ✓✓✓ | ✓ |
| 2–3 | Land use data | Origins and possible destinations, simulation of bicyclist behaviour, and safety analysis | ✓ | ✓✓✓ | ✓ |
| 2–3 | Digital Elevation Model or Slope map | Built environment suitability analysis and simulation of bicyclist behaviour | ✓ | ✓✓✓ | ✓ |
| 2–3 | Google map places, Point of Interest | Origins and possible destinations, built environment suitability analysis, simulation of bicyclist behaviour, and safety analysis | ✓ | ✓✓✓ | ✓ |
| 3 | GTFS, public transport data | Travel mode choice | ✓ | ✓ | ✓ |
| 3 | Tree Canopy Density – Vegetation | Built environment suitability analysis and travel mode choice | ✓ | ✓ | ✓ |
| 3 | Road network data: PSMA-Street network line or Open Street Map | Built environment suitability analysis, simulation of bicyclist behaviour, and safety analysis | ✓✓ | ✓✓✓ | ✓✓ |
| 3 | Shared bike data | Route choice analysis, estimation of future demand, and simulation of bicyclist behaviour | ✓✓✓✓ | ✓ | ✓ |
| 3 | Bicycling applications data: iBike, Strava, RiderLog, MapMyRIde | Origin-Destination, route choice analysis, and simulation of bicyclist behaviour | ✓ | ✓✓ | ✓ |

**Table 3.** *Cont.*

| Evolutionary Phase | Dataset | Possible Applications | Smart City and Infrastructure Cluster | Built Environment and Decision Making | People and Safety |
|---|---|---|---|---|---|
| 3 | Social media such as Twitter and Facebook | Bicyclist attitudes | ✓ | ✓ | ✓✓ |

Description: ✓ denotes 0–25% usage in the cluster. ✓✓ denotes 25–50% usage in the cluster. ✓✓✓ denotes 50–75% usage in the cluster. ✓✓✓✓ denotes more than 75% usage in the cluster.

*4.2. Evolutionary Trends*

The result of the temporal analysis shows three evolutionary phases in using data-driven approaches in bicycling planning from 1990 to 2021 in the reviewed studies (refer to Table 2). The first phase is from 1990 to 2000, focusing mainly on the people, demographics, and safety from accidents. In this phase, authors usually tried to use demographics and accident data to propose a framework to increase bicycling safety and understand bicyclists' behaviour using mathematical and regression modelling methods. The importance of safety and accident injuries continues to grow, and it was still a significant focus of the current research in 2021. In addition, this time slot is the start of a growing trend in using GIS in bicycling planning process [67], such as behavioural research for bicycle route selection (e.g., [68]).

In the second phase, from 2000 to 2009, researchers began to investigate using more sophisticated techniques. The temporal analysis results show that after 2000, research on bicyclists' behaviour in built environments and bicyclists' route selection models gained traction in the academic literature. There is also a growing interest in evaluating built environment form and function with GIS. Another trend in this period was using environmental audit tools to understand and quantify the sustainability benefits of active transport, such as walking and bicycling Level of Service (LOS) (e.g., [69]).

The research emerged in the third phase from 2010 to 2021 in the reviewed studies, deemed to deepen the previous topics using new technologies such as AI, AR, IoTs, or GPS to understand travel behaviour in urban transportation. With many cities implementing shared bike systems in 2010, such as Barcelona, Washington, Philadelphia, and Taipei, shared and public bike planning studies started to grow [70]. With the data collected through these systems, the opportunity to apply advanced data mining techniques to support bicycling planning has brought about the next generation of research. With the increasing collection of data from smartphone applications, researchers can access finer-scale geographical and temporal analysis of bicyclist behaviour. In recent years, we have witnessed the rise of machine learning [31] and deep learning techniques (e.g., [71]) for modelling, estimating, and predicting bicycle trips. Also, simulating bicyclists' behaviour using Agent-Based Modelling (ABM) techniques is an ongoing trend to support decision-making (e.g., [24]).

Overall, the research tends to change from macro to micro in scale. For example, on a national scale, Hunter & Huang, 1995 tried to understand how people would respond if we built a bicycling facility. On a small scale, researchers in a study by [72] used the IoTs in bike power to generate electricity. Also, research trends changed from more theoretical research, like recommendations for using helmets [73], to more applied and technical research, such as developing instrumented bicycle applications and websites to advance bicycling activities in detecting potential hazards [74]. The evolving nature of the data-driven approaches outlined in this systematic literature review has supported studies leading to more innovative bicycling planning methodologies, which are in turn facilitating more evidence-based approaches to bicycling planning in cities.

This section reviewed primary knowledge domain categories and the evolution of the research in the context of methods and themes. To the best of our knowledge, this review is the first to have an extensive systematic review in this area and provide an overall view of the trends and concepts. This review illustrates an evolution of data-driven approaches

from standard statical techniques to more machine learning analytics and advanced digital tools to support bicycling planning.

## 5. Conclusions and Future Research

Despite the awareness of the benefits of bicycling on health, the environment, and the economy, the number of trips made by bike is too small in most countries worldwide. While urban planners and policymakers are pivotal to making cities more inviting for bicycling, it is a significant challenge for them as they deal with cities with complex issues and numerous stakeholders across government and industry. The rapid development of new technology has helped planners to improve urban planning processes [75,76]. To enable planners to make effective decisions, finding out how, when, and why they should invest in the bicycling infrastructure is essential. The availability of the new data enables planners in this regard by providing access to more detailed information on bicycling behaviour in built environments. Nonetheless, accessing high-quality granular data on various aspects of bicycle movements across cities has limited the planners' ability to use them in the planning practice [4].

In this review, we have conducted a systematic review of the bicycling planning literature to understand the trends in using data and technology, specifically in land use and transport planning. To begin, the scope of the review is specified by conceptualising the topic. According to the defined keywords, an extensive search was conducted to summarise and examine the bicycling research from 1990 to 2021. This review investigates temporal and cluster analysis to understand how data-driven approaches have evolved and matured over time. Some researchers have previously examined the evolution of bike-sharing systems studies [17,27]; Yet, to the best of our knowledge, there are no published studies that comprehensively review the scope of research undertaken in the broader context of data-driven approach in bicycling planning.

The results from the systematic review show the evolutionary trend of using data-driven approaches in bicycling planning in three phases. The first phase, from 1990 to 2000, focused on people and safety issues. From 2000 to 2010, there was an effort to understand how the built environment affects active transport and how it can be applied in the planning and decision-making process in the second phase. From 2010 to 2021, in the third phase, the research topics were artificial intelligence, bicycling sharing systems, augmented reality, the internet of things, big and open data, data sharing, and smart cities.

The cluster analysis results show three main clusters of research associated with applying data-driven approaches to bicycling planning: (1) smart city and infrastructure, (2) built environment and decision-making, and (3) people and safety. In terms of knowledge domains, each cluster is further categorised into two knowledge domains of smart city and virtual reality, bicycling infrastructure and bike-sharing, built environment and travel behaviour, decision making and planning, people and demographics, and safety and accident prevention.

With the systematic review of the data-driven approaches in bicycling planning over the past thirty years, we identified a few gaps in the current literature and highlighted new directions for future research. With respect to notable gaps, a large number of studies focused on developing data-driven tools and methods for bicycling planning. Decision support systems are such a tool for more logical and scientific decision-making. However, the main point here is how to apply these tools to the practice of planning and designing. In addition, developing and utilising these tools and methods requires specialised skills and specialties that can be considered a barrier to their application in the practice of planning.

According to our findings, although many studies focus on people and their safety issues, only a few studies consider how people should be involved in the planning process. Therefore, excluding people from the process of planning is another major challenge in current urban governance and the decision-making process, especially in bicycling planning. Furthermore, there is a need to engage stakeholders and experts in the decision-making processes to explore their ideas [77].

The next possible direction can be improving urban planning tools such as GIS or decision support systems to enable effective collaboration of various parties with more sophisticated methodologies to evaluate and simulate urban environment systems more precisely. The data used in this review is limited to the SCOPUS database. Although SCOPUS includes a wide range of academic documents, some documents included in other datasets may be neglected. In addition, author co-citation network and journal co-citation is not considered in this review because of the large number of included studies. Future studies can reconsider this framework using other literature databases and more integrative analysis methods.

In sum, As the world of technology changes, planning methods should change with it. The potential to use data-driven methods in planning is strengthened, and more automated tools are used in decision-making. Thus, it is essential to provide an environment where planners can use these technologies in a collaborative environment with local people, stakeholders, information technologists, engineers, and scientists. Researchers and practitioners should develop and, importantly, use new tools and platforms to maximise the cooperation of these people while minimising the conflicts in their discussions and decision-making for the future plans of the cities.

This study provided valuable knowledge and information regarding the data-driven methods in bicycling planning for researchers, practitioners, and decision-makers by presenting an in-depth understanding of the methods used. In particular, core knowledge domains and evolutionary trends enable us to understand the research status around the world. Summarising primary datasets, methods, and tools used in bicycling planning presents a clear, valuable, and crucial guide for follow-up research and practices, which can be a significant foundation for developing data-driven methods in bicycling planning elsewhere.

**Supplementary Materials:** The following are available online at https://www.mdpi.com/article/10.3390/su142316319/s1, PRISMA guidelines [8].

**Author Contributions:** Conceptualization, P.Z., C.P., S.L., O.G.; methodology, P.Z., C.P., S.L., O.G.; software, P.Z.; validation, P.Z.; formal analysis, P.Z.; investigation, P.Z.; data curation, P.Z.; writing—original draft preparation, P.Z.; writing—review and editing, P.Z., C.P., S.L., O.G.; visualization, P.Z., S.L.; supervision, C.P., S.L., O.G.; All authors have read and agreed to the published version of the manuscript.

**Funding:** This research received no external funding.

**Institutional Review Board Statement:** Not applicable.

**Informed Consent Statement:** Not applicable.

**Conflicts of Interest:** The authors declare no conflict of interest.

### Appendix A  Key Concepts Definitions in Data-Driven Approaches

**Table A1.** Summary of key concepts in using data-driven approaches in the bicycling planning literature.

| Data-Driven and Technology Concepts | Definition |
| --- | --- |
| Smart city, digital city, virtual city | "A smart city is a well-defined geographical area in which high technologies such as ICT, logistic, energy production cooperate to create benefits for citizens." [78]. |
| Internet of Things (IoTs) | "An open and comprehensive network of intelligent objects that have the capacity to auto-organise, share information, data and resources, reacting and acting in the face of situations and changes in the environment." [79]. |
| Artificial Intelligence (A.I) | "The science of building intelligent agents to perform tasks like a human being." [80]. |

**Table A1.** *Cont.*

| Data-Driven and Technology Concepts | Definition |
|---|---|
| Cloud Computing | "It embraces cyber-infrastructure and builds on virtualisation, distributed computing, grid computing, utility computing, networking, and Web and software services." [81]. |
| Information and communication technologies (ICT) | "ICTs can refer to the wide range of computerised information and communication technologies." [82]. |
| Data-driven approaches (including big data, crowd-sourced data, open data) | "A data-driven approach can be defined as one whereby data availability is the central criterion for indicator development and data is provided for all selected indicators." [83]. |
| Intelligent Operation Centre (IOC) | "IOC provides measuring monitoring, and modelling facilities that integrate current systems into one solution to improve operational efficiency, planning, and coordination." [84]. |
| Building Information Modelling (BIM) | "A BIM is a digital representation of physical and functional characteristics of a facility. As such, it serves as a shared knowledge resource for information about a facility." [85]. |
| Digital twin | "Digital twins, a digital replica of city infrastructure linked to real-time city data, that envisioned to improve city monitoring, control, and decision making." [86]. |
| Virtual Reality (VR), augmented reality, VRGIS | " VR is a computer-generated simulation of a three-dimensional image or environment that can be interacted with in a seemingly real or physical way." [87]. |
| Living laboratory, experimental city, laboratory experiment | "Urban Living Laboratory are forums 'for innovation, applied to the development of new products and processes to integrate people into the entire development process, to explore and evaluate new ideas." [88]. |
| Digital participatory planning, Participatory Geographic Information Systems (GIS)., people GIS, collaborative GIS, geocollaboration | "Participatory GIS involves local communities in the creation of information to be fed into the GIS." [89]. |
| Smart governance, e-government, Planning support systems, decision support systems | "smart governance is the capacity of employing intelligent and adaptive acts and activities of looking after and making decisions about something." [90]. |

**Appendix B  Inclusion and Exclusion Criteria in the Search Engine**

((TITLE (bicycle OR bike OR bicycling) AND TITLE-ABS-KEY (gis AND collaboration) OR TITLE-ABS-KEY (digital AND participation) OR TITLE-ABS-KEY (digital AND people) OR TITLE-ABS-KEY (digital AND "decision making") OR TITLE-ABS-KEY (data OR information AND "decision making") OR TITLE-ABS-KEY (tool AND "decision making") OR TITLE-ABS-KEY (digital AND "built environment") OR TITLE-ABS-KEY (data OR information AND "built environment") OR TITLE-ABS-KEY (data OR information AND people) OR TITLE-ABS-KEY (data OR information AND community) OR TITLE-ABS-KEY (digital AND community) OR TITLE-ABS-KEY (gis AND community) OR TITLE-ABS-KEY (digital AND "user experience") OR TITLE-ABS-KEY (digital AND "public experience") OR TITLE-ABS-KEY (digital AND "people experience") OR TITLE-ABS-KEY (gis AND "experience") OR TITLE-ABS-KEY (gis AND "experience") OR TITLE-ABS-KEY (gis AND "user experience") OR TITLE-ABS-KEY (gis AND people) OR TITLE-ABS-KEY (gis AND government) OR TITLE-ABS-KEY (gis AND community) OR TITLE-ABS-KEY ("Smart city" OR "digital city" OR "virtual city" OR "IoT" OR "Internet of things" OR "Virtual reality" OR "geographic information system" OR "gis" OR "VR" OR "augmented reality" OR "virtual" OR "VRGIS" OR "AI" OR "Artificial Intelligence" OR "Cloud Computing" OR "agent based modelling" OR "abm" OR "ICT" OR "Information and communication technologies" OR "Data driven" OR "data-driven" OR "big data" OR "crowd-sourced data" OR "open data" OR " BIM" OR "Building Information Modelling" OR "Digital twin"

OR "interactive screens" OR "Community integrated GIS" OR "Participatory GIS" OR "collaborative GIS" OR "geocolaboration" OR "people GIS" OR "Community integrated geographic information" OR "Participatory geographic information" OR "collaborative geographic information" OR "people geographic information" OR "Smart government" OR "e-government" OR "digital government" OR "electric government" OR "laboratory experiment" OR "lab experiment" OR "planning support systems" OR "decision support systems" OR "pss" OR "dss" OR "living laboratory" OR "Laboratory governance" OR "experimental governance" OR "experimental city" OR "experimental city" OR "living lab" OR "user experience" OR "ambient intelligence" OR "e-planning" OR "digital planning"))) AND (LIMIT-TO (SUBJAREA, "COMP") OR LIMIT-TO (SUBJAREA, "ENGI") OR LIMIT-TO (SUBJAREA, "SOCI") OR LIMIT-TO (SUBJAREA, "MATH") OR LIMIT-TO (SUBJAREA, "ENVI") OR LIMIT-TO (SUBJAREA, "DECI") OR LIMIT-TO (SUBJAREA, "ENER") OR LIMIT-TO (SUBJAREA, "BUSI") OR LIMIT-TO (SUBJAREA, "ARTS") OR LIMIT-TO (SUBJAREA, "PSYC") OR LIMIT-TO (SUBJAREA, "ECON") OR LIMIT-TO (SUBJAREA, "HEAL") OR LIMIT-TO (SUBJAREA, "MUL T")) AND (LIMIT-TO (LANGUAGE, "English")).

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
