# Peer review of "Digital Bicycling Planning: A Systematic Literature Review of Data-Driven Approaches"

_sustainability, doi:10.3390/su142316319_

Round 1

Reviewer 1 Report

The Authors attempt to provide an overview of existing approaches to planning bicycling. The systematic literature review is followed by a short discussion. The paper distinguishes 3 evolutionary stages and 6 knowledge domains in using data-driven approaches for planning bicycling.

I would recommend considering the paper https://doi.org/10.1080/13683500.2021.2011841 as one of the recent data-driven approaches. Additionally, the use of cargo bikes should be addressed in the literature review (see https://doi.org/10.3390/en14144132, and  https://doi.org/10.3390/en14040839).

The references to papers should be formatted in accordance with the template recommended by the Sustainability journal.

Author Response

Thank you for your constructive comments. Please consider the following changes according to your comments:

  • These two topics have been added to the paper. Please refer to the Knowledge domains in the Discussion
  • some changes have been made to the conclusion as well

Reviewer 2 Report

This article is a great systematic review paper in terms of data-driven approaches to planning for bicycling. It is interesting for potential readers and can be considered for publication if the following concerns can be addressed properly.

1. Keywords of the study appeared twice. One is directly after the abstract, please make corrections.

2. Table1 depicts three phases in bicycling planning based on the co-occurrence of keywords, which is great. However, it is a bit hard to follow the content in the table. Please redesign the table and make it more readable.

3. In line 240 you mentioned “The first cluster is smart city and infrastructure, divided into two sub-categories 1) smart city and virtual reality and 2) bicycling infrastructure and bike-sharing systems.” And in the next paragraph (line 249), you mentioned “The second sub-category is infrastructure and bike-sharing systems” again. The structure and presentation can be somewhat misleading. 

Author Response

Thank you for your constructive comments. Please consider the following changes according to your comments:

  1. It is removed
  2. It is reformatted and presented as an image to make it more readable.
  3. It is edited accordingly

Reviewer 3 Report

The paper is about an interesting vast literary review, discussed with clear methodology. We suggest to improve the quality by taking care of some aspects.

In the title the word "techology" should be better precised (i.e. digital) as it may refer to many different kinds.

There are many quoting from the literary review that are not correctly numbered in relation to the references.

In the conclusions, the concept espressed in lines 420-422 should be better precised.

The findings of the work could be enhanced by better structuring the conclusions. We suggest to report more critical comments and indications for properly using the results, foreseeing the cultural perspective. 

Author Response

Thank you for your constructive comments. Please consider the following changes according to your comments:

  1. It is changed to digital bicycling planning.
  2. It is fixed.
  3. The conclusion has been restructured, and some other points have been added to show the results better.

Reviewer 4 Report

The problem of cycling planning perfectly fits into the strategy of sustainable development of transport. Many strategies of various entities include concepts of its development. Investments are realised in this area, supported by various funds dedicated to sustainable development of transport.

The authors' research presented in the reviewed article 'Technology and bicycling planning: A Systematic Literature Review of Data-Driven Approaches' systematises and organises the emerging concepts in the literature in this area.

I was pleased to read its content. It is clearly and logically written. The article has a clear purpose. Both the overview of the concepts, the method and the results can be useful to theoreticians for writing their scientific papers as well as to practitioners, because on the basis of the analysis the authors point to concrete solutions for cycling planning.

Author Response

We appreciate your constructive comments!